# Simultaneous Occurrence of Multiple Neoplasms in Children with Cancer Predisposition Syndromes: Collaborating with Abnormal Genes

**DOI:** 10.3390/genes14091670

**Published:** 2023-08-24

**Authors:** Gabriela Telman, Ewa Strauss, Patrycja Sosnowska-Sienkiewicz, Magdalena Halasz, Danuta Januszkiewicz-Lewandowska

**Affiliations:** 1Department of Pediatric Oncology, Hematology and Transplantology, Poznan University of Medical Sciences, Szpitalna 27/33, 60-572 Poznan, Poland; gabriela.telman@gmail.com (G.T.);; 2Institute of Human Genetics, Polish Academy of Sciences, Strzeszynska 32, 60-479 Poznan, Poland; strauss@man.poznan.pl; 3Department of Pediatric Surgery, Traumatology and Urology, Poznan University of Medical Sciences, Szpitalna 27/33, 60-572 Poznan, Poland; sosnowska@ump.edu.pl

**Keywords:** cancer predisposition syndromes, genetic abnormalities, neoplasms, simultaneous occurrence, pediatric cancers, personalized treatment, tumor suppressor genes

## Abstract

The identification of cancer predisposition syndromes (CPSs) plays a crucial role in understanding the etiology of pediatric cancers. CPSs are genetic mutations that increase the risk of developing cancer at an earlier age compared to the risk for the general population. This article aims to provide a comprehensive analysis of three unique cases involving pediatric patients with CPS who were diagnosed with multiple simultaneous or metachronous cancers. The first case involves a child with embryonal rhabdomyosarcoma, nephroblastoma, glioma, and subsequent medulloblastoma. Genetic analysis identified two pathogenic variants in the *BRCA2* gene. The second case involves a child with alveolar rhabdomyosarcoma, juvenile xanthogranuloma, gliomas, and subsequent JMML/MDS/MPS. A pathogenic variant in the *NF1* gene was identified. The third case involves a child with pleuropulmonary blastoma and pediatric cystic nephroma/nephroblastoma, in whom a pathogenic variant in the *DICER1* gene was identified. Multiple simultaneous and metachronous cancers in pediatric patients with CPSs are a rare but significant phenomenon. Comprehensive analysis and genetic testing play significant roles in understanding the underlying mechanisms and guiding treatment strategies for these unique cases. Early detection and targeted interventions are important for improving outcomes in these individuals.

## 1. Introduction

Cancer incidence is increasing in all age groups in the general population, but neoplasms in children and adolescents are still considered rare diseases [1,2,3]. The global incidence of childhood cancer is estimated to be nearly 400,000 cases per year [4]. Multiple diagnoses are extremely rare. While the exact causes of oncological diseases remain largely unknown, the role of genetic factors and the identification of syndromes that predispose individuals to cancer are gradually being identified and understood with advancements in genetics and genomics [5,6,7,8,9]. With medical advancements, targeted treatments, anticipatory screening, and the prevention of subsequent cancers are possible, which is especially important for individuals with cancer predisposition syndromes (CPSs) [10,11,12,13]. Despite widespread prevention efforts, the early detection of cancer is not fully effective.

As early as the 19th century, it was suggested that the possibility of developing a second cancer should not be excluded, even if the first cancer has been completely removed through surgery [14]. An increased risk of developing a second primary cancer among cancer survivors is well-known, but the simultaneous occurrence of multiple cancers remains unclear. Depending on the timing of the onset of multiple cancers, they can be classified as concurrent, synchronous, or metachronous. Most authors consider a time frame of two months or less for calling a cancer synchronous. For simultaneous cancers, some authors refer to them as a subset of lesions or use the term “metachronous” cancer [14].

Given the rarity of CPS, the main aim of this article is to identify cases with concomitant or metachronous multiple neoplasms in a Polish single-center study and to present a comprehensive genetic, phenotypic and a clinical analysis on disease progression and treatment.

## 2. Materials and Methods

### 2.1. Study Population

Out of the 2387 patients who were newly diagnosed with cancer and hospitalized at the Karol Jonscher Clinical Hospital of the Karol Marcinkowski Medical University in Poznan between 2000 and 2021, 182 were identified as having CPSs. Cancer predisposing syndromes were defined based on the presence of characteristic phenotypic abnormalities, such as Down syndrome, neurofibromatosis type 1 (NF1), Beckwith–Wiedemann syndrome (BWS), Fanconi anemia, tuberous sclerosis complex (TSC), Turner syndrome, dyskeratosis congenita, and others, or through confirmation of mutations in genetic tests. In the case of Down syndrome, the parents and the child were consulted in each case by a specialist geneticist, who confirmed the syndrome with karyotype testing from peripheral blood lymphocyte cultures. Qualification for genetic testing was carried out based on consultation with a clinical geneticist specialist. The patient was referred to a genetic clinic in specific situations; where the family history of first- or second-degree relatives was burdened with cancer, in the case of developing cancer at an infant age, or in situations where histopathological diagnosis indicated a high likelihood of a CPS, such as pleuropulmonary blastoma, adrenal cortex carcinoma, infantile myofibromatosis, or retinoblastoma. Among them, only three cases of simultaneous or metachronous multiple cancers were recognized.

### 2.2. Molecular Analysis

For genotype analysis, genomic DNA was extracted from circulating blood lymphocytes using the QIAamp DNA Kit (Qiagen GmbH, Hilden, Germany), following the manufacturer’s instructions. Targeted next-generation sequencing (NGS) was utilized for the identification of pathogenic variants (performed at the Medical University of Lodz, CeGaT’s Diagnostics, GENOMED, and Blueprint Genetics). NGS using the TriSight One Expanded Sequencing Panel by Illumina allows the analysis of 6699 genes with documented clinical significance. The sequencing, consisting of 298 cycles, was performed on the NextSeq 550 instrument by Illumina. The minimal read depth for the individual sequence was 30× and was achieved for 99% of the examined genes. The final analysis of the oncology gene subpanel was conducted using Variant Studio v.3.0 (Illumina Inc., San Diego, CA, USA), Agilent SureCall v.4.1 (Agilent Technologies; Santa Clara, CA, USA), and IGV v.2.3 (Broad Institute, Cambridge, MA, USA) software. Bioinformatics prediction was carried out using Mutation Taster, SIFT, and PolyPhen platforms. Pathogenicity classifications were determined based on ACMG guidelines. It is worth noting that this method does not allow for the detection of copy number variations (CNV). The examined oncology gene subpanel includes 130 genes. A detailed list of genes in the oncology panel is given in Appendix A. Flow cytometry analysis was conducted to assess the immunophenotype of bone marrow cells using GUAVA EasyCyte 6HT-2L flow cytometer at Poznan University of Medical Sciences Core Facility. Fluorescence in situ hybridization (FISH) analysis was performed to analyze chromosome number aberrations at the Department and Clinic of Hematology and Bone Marrow Transplantation, Poznan University of Medical Sciences. Additional molecular diagnostic analyses included the assessment of loss of heterozygosity (LOH) of the *NF1* gene, and the analysis of somatic mutations associated with juvenile myelomonocytic leukemia (JMML) conducted by The European Working Group of Myelodysplastic Syndrome in children at the University of Freiburg. Chromosome 11 methylation abnormality tests and analysis of deletions and duplications in the 11p15 region using the Multiplex Ligation-dependent Probe Amplification (MLPA) method were performed at Centogene GmbH, Germany. Blood fetal hemoglobin (HbF) levels were measured using the high-performance liquid chromatography (HPLC) method at Diagnostyka company in Poznan, Poland.

### 2.3. Cytological and Histological Analysis

Smear analyses were also performed using peripheral blood and bone marrow aspirate smear tests. For this purpose, the May–Grunwald–Giemsa method of staining was used. All the slides were analyzed and reported by a hematology specialist. All the diagnostic core biopsies of tumors were assessed using formalin-fixed, paraffin-embedded (FFPE) tissue sections stained with Hematoxylin and Eosin (H&E).

### 2.4. Data Visualization

The diagrams were prepared using Creately software, available at URL: https://creately.com (accessed on 26 June 2023).

## 3. Results

### 3.1. Identification of Cases with Simultaneous/Metachronous Multiple Cancers in a Studied Cohort

Out of the 2387 patients who were newly diagnosed with cancer, 182 were identified as having CPSs, which represents 7.6%. This group included 25 patients with Down syndrome, 49 with neurofibromatosis type 1-NF1, 2 with neurofibromatosis type 2-NF2, 2 with multiple endocrine neoplasia type 1-MEN1, 14 with Beckwith–Wiedemann syndrome BWS or isolated hemihyperplasia syndrome, 1 with Wilms tumor-aniridia syndrome-WAGR syndrome, 1 with Wiskott–Aldrich syndrome-WAS, 20 with Li–Fraumeni syndrome-LFS, 2 with Von Hippel–Lindau (VHL), 2 with Noonan-like syndrome, 1 with tyrosinemia type 1 (FAH), 1 with trisomy 18 (Edwards syndrome), 1 with tuberous sclerosis complex (TSC), 3 with Turner syndrome, and 1 with dyskeratosis congenita. All the children with Down syndrome were found to have trisomy of chromosome 21 on karyotype testing. The remaining children presented mutations in the following genes (number of cases in parentheses): *ALK* (1), *ATM* (1), *BRCA1/2* (7), *CDKN2A* (2), *CHEK2* (5), *DICER1* (3), *FACC* (4), *FAP* (1), *FGFR3* (1), *GATA2* (3), *KIT* (2), *MSH2* (2), *MSH6* (2), *NBN* (2), *PDGFRA* (1), *PDGFRB* (1), *PHOX2B* (2), *PIK3* (1), *PMS2* (2), *PTEN* (1), *PTPN11* (1), *RB1* (5), *RET* (2), *SMARCA4* (1), *SMARCB1* (1), *TSC1* (2), and *WT1* (1). Despite the relatively large number of patients included in the analysis, only three children were found to have simultaneous or metachronous multiple cancers, which are described in detail. A diagram showing the analysis and workflow is presented in Figure 1.

### 3.2. Pathogenic Variants of the BRCA2 Gene—Case 1

The first patient is the firstborn child of a mother with hypothyroidism. The parents are unrelated, and the father has no history of chronic diseases. There is no family history of cancer. The girl was born at full term (at 40 weeks’ gestation) with a birth weight of 2950 g and scored 10 on the Apgar scale. Soon after birth, extensive café-au-lait spots were observed, primarily on the right thigh, which resembled those seen in the mother. Starting from the second month of life, the child exhibited left hemiplegic hypertrophy (Figure 2) and a slight delay in psychomotor development.

At 7 months of age, the child was admitted to hospital due to isolated swelling of the left upper limb. During this hospitalization, additional abnormalities were detected, including a horseshoe kidney and an accessory spleen. At 14 months of age, the parents noticed a nodule in the child’s left scapular region, which was subsequently diagnosed as embryonal rhabdomyosarcoma (ERMS) following its removal. On ultrasound, this tumor measured 2.5 cm × 1.8 cm × 1.0 cm and showed a probable connection to the muscle. Further tests revealed a horseshoe-shaped tumor in the kidney, measuring 8.0 cm × 5.7 cm × 8.7 cm, as identified by abdominal ultrasound, and confirmed by computed tomography (CT) scan. The biopsy results confirmed it to be a nephroblastoma (Wilms tumor, WT) of mixed, intermediate-risk, G3 type. Furthermore, an optic nerve glioma was detected during magnetic resonance imaging (MRI). To validate the histopathological findings, multiple diagnostic centers in Poland (The Children’s Memorial Health Institute in Warsaw) and Germany (at the University of Bonn and the Heidelberg University) were consulted, all of which confirmed the diagnoses of ERMS and Wilms tumor.

Given the concurrence of these three cancers, additional NGS testing was conducted. This revealed two distinct heterozygous pathogenic variants in the *BRCA2* gene, namely c.1773_1776delTTAT (p.Ile591MetfsTer22; dbSNP rs80359304) inherited from the mother, and 886delGT(c.658_659delGT; p.Val220fs; dbSNP rs80359604) inherited from the father.

As a result of the hemihypertrophy, chromosome 11 methylation abnormality tests and chromosome 11 MLPA were conducted, which did not identify any large deletions or duplications in the 11p15 region. Utilizing Illumina TruSight One Expanded Sequencing Panel and Illumina NexSeq 550 instrument (Illumina Inc., San Diego, CA, USA), NGS analysis was performed, focusing on genes with documented clinical relevance to oncologic disorders. The analysis, conducted using Variant Studio v.3.0 and IGV v.2.3 software, did not reveal the presence of any other pathogenic changes. Additionally, imprinted DNA methylation analysis was carried out at various loci, including *DIRAS3* (1p31); *PLAGL1* (6q24); *GRB10* (7p12); *PEG1*/*MEST* (7q32); *KCNQ1OT1/H19/IGF2 DMRO* (11p15); *DLK1* (14q32); *SNRPN* (15q11); *PEG3* (19q32); and *NESPAS/GNAS* (20q13). Chemotherapy was initiated based on the Cooperative Weichteilsarkomstudiengruppe (CWS) protocol and, following the confirmation of Wilms tumor through histopathological examination, the treatment continued according to the UMBRELLA protocol of the International Society of Pediatric Oncology Renal Tumor Study Group (SIOP-RTSG). Two and a half years after the original diagnosis, an MRI revealed the presence of a medulloblastoma tumor (classic type, central nervous system (CNS) WHO G4) in the left cerebellar hemisphere. Considering the outcome of the non-radical surgery and tumor progression during chemotherapy, the child necessitated not only surgical intervention but also chemotherapy and radiation therapy. The presence of *BRCA2* pathogenic variants posed a risk of Fanconi anemia. Anticipating bone marrow aplasia due to the treatment, anticipatory allogeneic hematopoietic stem cell transplantation (HSCT) was performed. Figure 3 provides a timeline of the patient’s disease progression.

### 3.3. Pathogenic Variant of the NF1 Gene—Case 2

The second patient is the first-born child of unrelated parents, both of whom are healthy with no family history of chronic diseases. However, each parent has more than 5 café-au-lait spots. The girl was born prematurely at 36 weeks’ gestation due to premature rupture of the fetal membranes, with a birth weight of 3315 g and an Apgar score of 10. Café-au-lait spots were initially observed on the patient’s skin after birth, and their number increased over time. The patient now has multiple café-au-lait spots distributed throughout the body. At 12 months of age, the patient’s mother noticed blood in her diaper, which prompted her to seek medical attention. An abdominal ultrasound revealed a pelvic tumor measuring over 10.0 cm in diameter. A subsequent MRI confirmed a vaginal tumor measuring 11.0 cm × 7.7 cm × 5.8 cm (Figure 4).

A histopathological examination of the biopsy led to a diagnosis of alveolar rhabdomyosarcoma (ARMS). Simultaneous MRI of the CNS revealed eight gliomas, including one involving the optic nerve, measuring up to 1.5 cm × 1.4 cm × 1.2 cm. Additionally, three yellow-brown skin lesions were found in the craniofacial region, which were confirmed to be Xanthogranuloma Juvenile through histopathological examination. The patient underwent treatment following the CWS protocol, which included chemotherapy and postponed radical surgery. After treatment, complete remission of ARMS was achieved, and the Xanthogranuloma Juvenile lesions also resolved. The results of NGS genetic analysis with a TruSight One Expanded Sequencing Panel revealed the presence of a heterozygous pathogenic mutation in the *NF1* gene (c.574C>T; p.Arg192Ter; dbSNP rs397514641). Neither the mother nor the father was a carrier of the change in the *NF1* gene, so mutation in the *NF1* gene in Case 2 was identified as a de novo mutation.

After 2.5 years from the initial diagnosis of malignancy, a follow-up blood examination showed thrombocytopenia, which progressed to bicytopenia, with the addition of transfusion-dependent anemia. Further studies were conducted on peripheral blood and bone marrow smears (Figure 5 and Figure 6, respectively).

Flow cytometry studies revealed the presence of immature myeloid cells with the immunophenotype CD33 + CD38 + CD31 + CD11b +/− CD11c + CD64 + CD19 − CD10 − CD13 + CD34 − CD117 – HLADR − CD20 − CD22 − CD3 − CD5 − CD7 − CD65 + CD15 + CD123 + MPO + TdT − FSC mid, SSC high. In the bone marrow, these cells accounted for 77% of the total cell population, with 13% of them being myeloblasts.

Several additional tests were performed. FISH analysis showed that 85% of the cells exhibited chromosome 5 monosomy. Further molecular analyses revealed *NF1* LOH but no somatic mutations typically associated with JMML. No mutations were found in *PTPN11*, *NRAS*, *KRAS*, *CBL*, *JAK2* exon 12, *CALR* exon 9, *MPL* exon 10, or *ASXL1* exon 13.

Although *NF1* mutations are risk factors for JMML, the presence of monosomy of chromosome 5, and low levels of fetal hemoglobin (HbF), were more indicative of secondary myelodysplastic/myeloproliferative syndrome (MDS/MPS). Following three courses of treatment with azacitidine, the patient underwent allogeneic hematopoietic stem cell transplantation (allo-HSCT). The disease course timeline for the second patient is depicted in Figure 7.

### 3.4. Pathogenic Variant of the DICER1 Gene—Case 3

The third patient is a second-born child. His parents are unrelated, with no family history of chronic diseases other than polycystic thyroid goiter (in the mother), which is also present in the child’s grandmother and aunt. He was born at 38 weeks’ gestation with a birth weight of 4220 g and a 10-point Apgar score. At the age of 6 months, he was admitted to our hospital due to suspected lung abscess. A CT scan revealed a 4.6 cm × 4.5 cm × 5.5 cm lesion in the right lung, and an ultrasound showed a 2.7 cm × 1.9 cm × 3.0 cm lithocystic lesion in the left kidney.

Considering the family history of multinodular thyroid goiter and the presence of a lung and kidney tumor in the boy, there was a suspicion of pleuropulmonary blastoma in the right lung and nephroma in the left kidney. Following the removal of the lung tumor, histopathological examination confirmed pleuropulmonary blastoma type II (PPB II). Due to the incomplete removal of the tumor during surgery (a non-radical surgery), chemotherapy was administered according to the Cooperative Weichteilsarkomstudiengruppe (CWS) protocol. After three months of chemotherapy, a follow-up abdominal ultrasound revealed progression of the left kidney tumor, leading to the decision to perform a nephrectomy. Histopathological examination confirmed pediatric cystic nephroma/nefroblastoma (Wilms tumor, WT). The boy continued chemotherapy as per the CWS protocol, and a resection of the residual lung tumor was also performed. The child successfully completed the treatment and remained in complete remission. NGS genetic analysis, using the same TruSight One Expanded Sequencing Panel, identified a pathogenic heterozygous variant in the *DICER1* gene (c.4930T>G, p.Leu1573Arg). An identical change in the *DICER1* gene was confirmed in the child’s mother (the father showed no changes in the genes tested). Genetic counseling was extended to the entire maternal family. Figure 8 illustrates the disease course timeline of the patient.

## 4. Discussion

As the incidence of cancer increases with age, cancers in the pediatric population remain in the group of rare diseases, and their multiple diagnoses, especially within 3 years, represent an even rarer phenomenon [15]. Although the nomenclature of multiple neoplasms seems clear, in each case there is at least one complexity. Due to the time each disease takes to develop to reach the picture presented in examination studies, it is not known whether the first lesion occurred within two months of the other when both are seen at the same time. Although malignant tumors imply a shorter expansion time, this still depends on individual and accurate diagnosis.

Depending on the type of mutation, the presence of a genetic background is associated with the onset of the disease (sometimes indicating a better or worse prognosis) or enables its prevention [5,8,11,16]. The clinical cases reported in this paper exemplify the links identified between specific mutations in the genome and the induction of cancers in infancy and early childhood (at 14, 12, and 6 months of age at diagnosis, respectively). The simultaneously diagnosed tumors in the first reported case were embryonal rhabdomyosarcoma (ERMS), nephroblastoma (Wilms’ tumor, WT), and glioma, which (after 2.5 years from the first oncological diagnosis) progressed to medulloblastoma classical CNS type WHO G4, which can be classified as a metachronous tumor. What is more, this individual suffered Fanconi anemia. This complex phenotype was diagnosed in a carrier of two different heterozygous variants in the *BRCA2* gene. These variants were previously associated with hereditary CPSs [17,18]. The second clinical case involved a carrier of the de novo *NF1* gene mutation who was concurrently diagnosed with alveolar rhabdomyosarcoma (ARMS), juvenile xanthogranuloma, and gliomas located in the central nervous system (CNS), including in the optic nerve. Approximately 2.5 years from the first oncological diagnosis, the metachronous disease—JMML/MDS/MPS—was found, but it remains in full remission of previous neoplasms. The observed variant of NF1 was previously associated several times with hereditary CPSs and, based on that, was assigned using Ambry Genetics^®^ General Variant Classification Scheme as a pathogenic. The third patient (with a history of polycystic thyroid goiter in the female line in his mother’s family members) was simultaneously diagnosed with pleumopulmonary blastoma type II (PPB II) and pediatric cystic nephroma/nefroblastoma related to mutations in the *DICER1* gene. After 1.5 years of follow-up, the child remains in full remission. The identified variant had been previously shown to be associated with pleuropulmonary blastoma [19] and may also be involved in other cancers [20]. No occurrence of cancer in the mother of the third child presented (despite carrying the same *DICER1* gene mutation) can be explained by the methodological limitations of the study. This may result from the analysis of only 130 genes from the oncology subpanel and not all the 6699 genes of TriSight One Sight One Expanded Sequencing Panel by Illumina with documented clinical significance. It should also be kept in mind that the possibility of gene copy number variations (CNV), that are impossible to determine by the technique used, can be revealed through a high-resolution whole-genome array.

The characterization of *BRCA2*, *NF1*, and *DICER1* genes, in terms of the function of the proteins they encode and their impact on tumorigenesis, is presented in Table 1.

Awareness of CPS, manifesting in increased susceptibility to cancer formation, should lead to adjustments in therapy, insofar as the proposed procedures may increase the possibility of neogenesis. However, the indications are very often ambiguous due to the benefit/risk ratio, which clinicians must keep in mind and consider. Avoiding high-risk courses of treatment can lead to the progression of the current cancer and may lead to the patient’s death before developing another cancer due to the procedure under consideration. These topics are indirectly related to medical futility [30], an awareness of which should be an ethical challenge for everyone, as it brings more suffering than long-term or even short-term benefit to the patient.

Given the current knowledge of CPS, which is augmented by case-studies, defining the standards for treatment modification is crucial for the transparency of medical procedures with oncology protocols. Priority must be placed upon the quality of life of the patients and their families, not only during but also after the treatment offered. Minimizing the risk of adverse events, with the development of another metachronous cancer as one of the most important side effects, can improve healthcare delivery [31,32,33]. Very often, patients with CPSs are at higher risk of developing several cancers, not just one type [1,16], so preventive management is essential to avoid increasing the already elevated risk of cancer in these cases. The question “How should we guide treatment?” remains open, as long as we discuss the very or even extremely rare genetic disorders that cause CPS. There is currently one well-known genetic disorder, Down syndrome, for which distinct diagnostic and treatment pathways have been developed, leading to early diagnosis and reduced mortality among these patients [5]. Such supportive care efforts should also be undertaken in other cases of CPS and lead to the establishment of dedicated procedures for such patients.

As suggested above, diagnostic and therapeutic management should not further increase the risk of developing secondary cancers in patients with CPSs. The radiation used not only in CT scans and other imaging procedures, but especially in therapy, should be avoided if proven to cause carcinogenesis, even when considering long-term risks measured in decades, not just a few years [34,35,36]. Moreover, an awareness of the implications of CPS should prompt therapists to be more vigilant in monitoring in order to detect cancers at an early stage. The lack of developed standards should motivate most specialist centers to introduce such management algorithms in the face of the growing number of patients with CPSs and the detection of new mutations that cause carcinogenesis. The validity of anticipatory treatment could also be considered, but further studies calculating the benefit/risk ratio would be needed to fully discuss and answer this thesis. So far, there is no targeted treatment, however, further dynamic development in this field will help in its generation.

## Figures and Tables

**Figure 1 genes-14-01670-f001:**
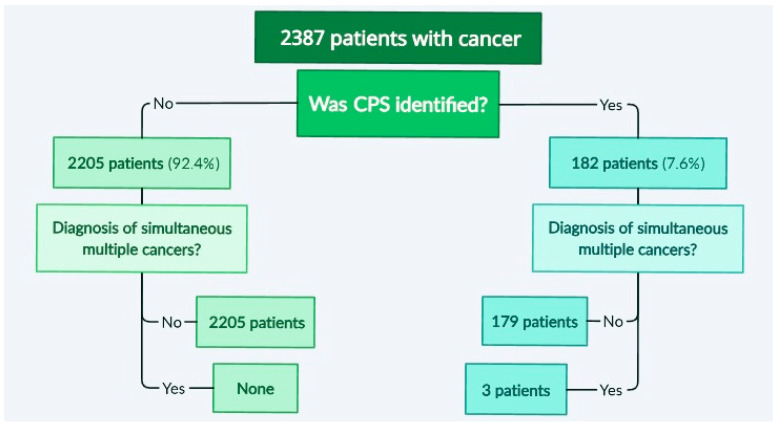
A diagram showing the analysis and workflow. CPS—cancer predisposition syndrome.

**Figure 2 genes-14-01670-f002:**
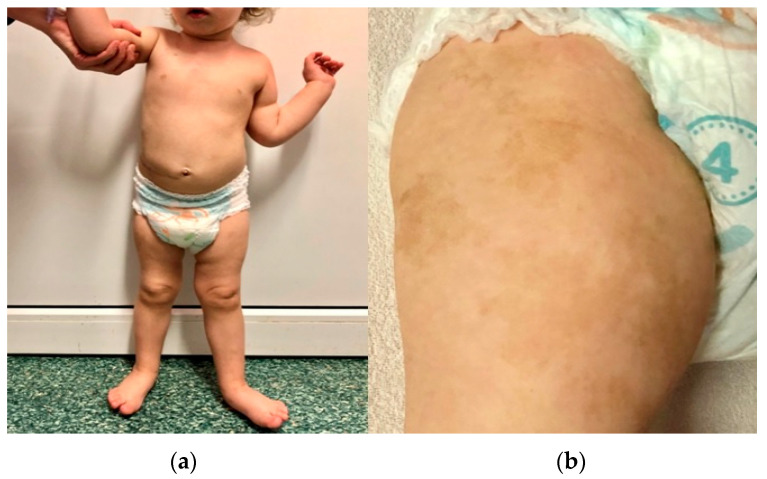
Photographs of the first patient, taken in January 2021, showing left-sided hemihypertrophy (**a**) and café-au-lait spots on the right thigh (**b**).

**Figure 3 genes-14-01670-f003:**
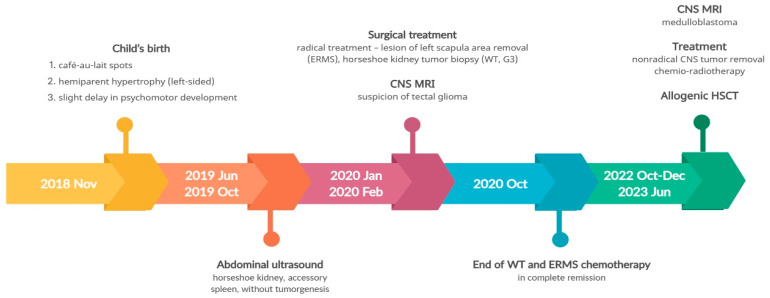
Timeline illustrating the disease course of Case 1.

**Figure 4 genes-14-01670-f004:**
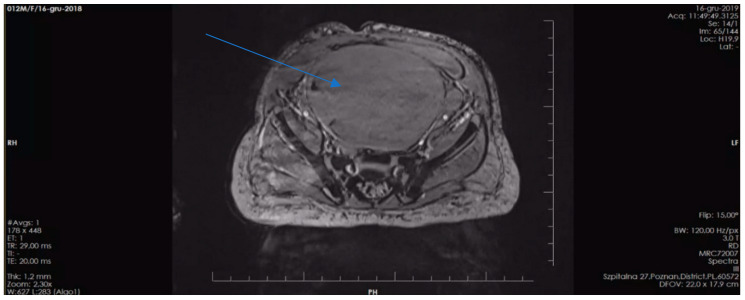
Pelvic MRI scan illustrating a large vaginal tumor (indicated by arrow) in the second patient presented.

**Figure 5 genes-14-01670-f005:**
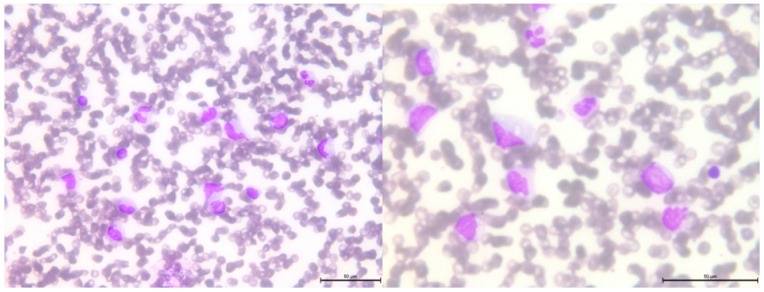
Peripheral blood smear showing leukocytosis with monocytosis, left-shifted neutrophils and immature cells, dysplastic granulocytes with pseudo Pelger cells and pycnotic nuclei, and 6% blasts.

**Figure 6 genes-14-01670-f006:**
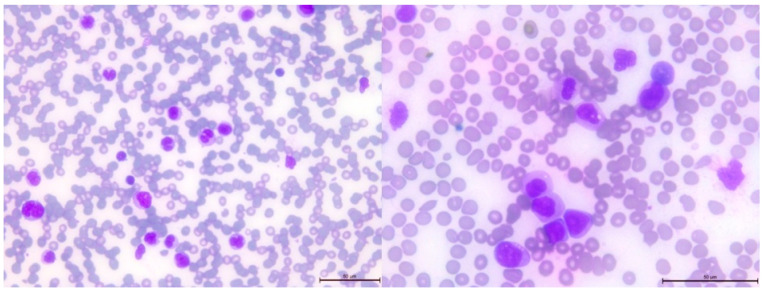
Bone marrow smear showing reduced cell content, absence of megakaryocytes, monocytosis with dysplastic and immature monocytes, very dysplastic myelopoiesis with pseudo Pelger cells, and aplastic erythropoiesis. Additionally, 13% blasts are observed.

**Figure 7 genes-14-01670-f007:**
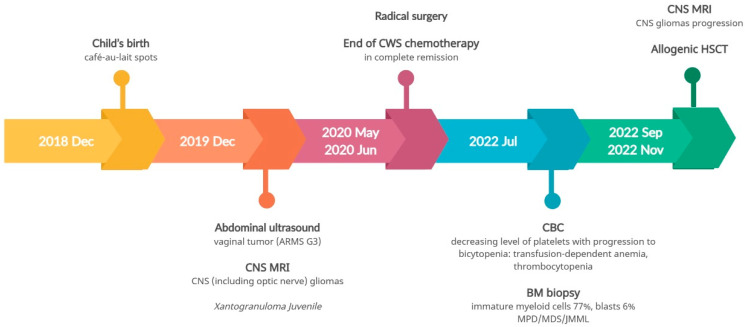
Timeline illustrating the disease course of Case 2.

**Figure 8 genes-14-01670-f008:**
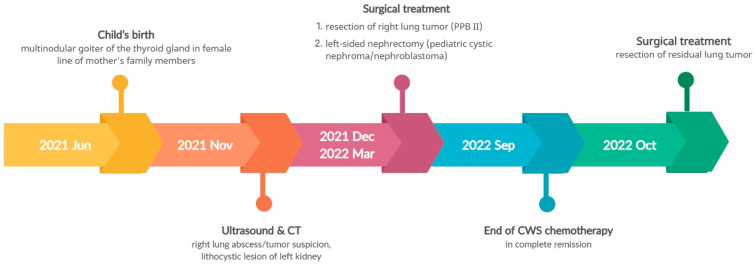
Timeline illustrating the disease course of Case 3.

**Table 1 genes-14-01670-t001:** Characterization of *BRCA2*, *NF1*, and *DICER1* genes in terms of the function of the proteins they encode and their impact on tumorigenesis (based on previous data [7,11,21,22,23,24,25,26,27,28,29]).

Gene	Function	Mutation Consequences	Health Conditions Related to Genetic Changes
** *BRCA2* **	Encodes the protein BRCA2 (BReast CAncer gene 2) This protein plays a crucial role in repairing DNA damage and functions as a tumor suppressor.	Most pathogenic variants of the *BRCA2* gene result in the production of an abnormally small and nonfunctional form of the BRCA2 protein from one copy of the gene in each cell. This leads to a reduced amount of available protein, which impairs efficient DNA repair. The accumulation of these defects can trigger uncontrolled cell growth and division, ultimately leading to carcinogenesis.	Breast cancer in men and women Ovarian/Prostate cancer Fanconi anemia Acute myeloid leukemia (AML) Acute lymphoblastic leukemia (ALL) Blastomas: nephroblastoma (Wilms tumor), hepatoblastoma, neuroblastoma, medulloblastoma Central nervous system (CNS) tumors Pancreatic cancer
** *NF1* **	Encodes the protein neurofibromin 1 This protein stimulates the GTPase activity of G proteins, including the proto-oncogene RAS, and acts as a tumor suppressor Plays a role in cAMP level regulation in astrocytes and Schwann cells Regulates melanosomes transport	Many pathogenic variants of the *NF1* gene lead to the production of an extremely short version of neurofibromin, which is unable to inhibit cell division. The presence of these variants may result in a constitutive activation of the signal transduction pathway, as the G proteins are constantly active due to the failure to degrade the attached GTP molecule. The loss of protein activity also increases susceptibility to harmful UV effects.	Neurofibromatosis type 1 Juvenile myelomonocytic leukemia Malignant peripheral nerve sheath tumors (MPNST) Central nervous system (CNS) gliomas (mainly optic nerve glioma) Pilocytic astrocytoma Gastrointestinal stromal tumor Pheochromocytoma
** *DICER1* **	Encodes the protein ribonuclease III, which contains an RNA helicase motif with a DEXH box in its amino terminus and an RNA motif in the carboxy terminus. Functions as a ribonuclease and is essential for the RNA interference and small temporal RNA (stRNA) pathways, producing the active small RNA component that represses gene expression and may initiate apoptosis. Also acts as a potent antiviral agent, exhibiting activity against RNA viruses, including the Zika and SARS-CoV-2 viruses. Alternative splicing leads to the formation of multiple transcript variants.	*DICER1* can be considered both a tumor suppressor gene, due to loss-of-function mutations, and an oncogene, due to gain-of-function mutations. It is believed to function as a haploinsufficient tumor suppressor gene, where the loss of one allele leads to tumor progression, but the loss of both alleles has an inhibitory effect on tumor development, implying that one intact allele is necessary for cell survival.	Multinodular goiter (MNG), thyroid cancer Blastomas: pleumopulmonary blastoma (PPB), pineoblastoma, medulloblastoma, pituinary blastoma, nephroblastoma (Wilms tumor) Cystic nephroma Rhabdomyosarcoma (RMS) Sertoli–Leydig cell tumor Hodgkin lymphoma

## Data Availability

Data available on request due to restrictions.

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
