# Peer review of "Simultaneous Occurrence of Multiple Neoplasms in Children with Cancer Predisposition Syndromes: Collaborating with Abnormal Genes"

_genes, 2023, doi:10.3390/genes14091670_

Round 1
Reviewer 1 Report
I am accepting current version.
NA.
Reviewer 2 Report
In the present work Telman et al. reported on different cases of pediatric patients with multiple neoplasms. The present work, overall, is interesting, yet it is not well-written nor scientifically sound. In particular:
First of all, the paper type does not constitute a research article, but rather a case report. The other thing that is importantly missing is the “patients and methods” section. Patient information and methodology are completely missing. The authors report on NGS, immunophenotyping, etc. with no reference on how these methods were concluded, performed, analyzed. No mention of data analysis, how were mutations identified, what was the molecule of reference; thus, this is as serious omission from the authors.
Thus, the authors performed a mere presentation of some cases (although rare) and just described the findings. Further on, genetic variation, by no means, predisposes to cancer from an aetiological point of view. Rather, neoplasms and oncogenesis have largely unknown mechanics, which is probably a mixture of infections (probably), genetic variation, gene regulation, chance and timing. Thus the authors should tune down their aetiological explanations on predispository syndromes and childhood neoplasms.
Overall, the present work does not have merit for publication as a research article, since it is a case study as well as there are important flaws in the rationale itself.
minor editing
Reviewer 3 Report
This paper is an investigation of abnormal gene activity in 3 cases of multiple primary neoplasms in a pediatric population. It is a reasonable demonstration of different gene abnormalities in these patients and provides useful information for further research.
The paper needs considerable polishing of the English, especially in the Discussion section. The abstract uses crucial 3 times. Another word could be chosen for one or two of these. Line 45,, is there a Reference for Bilroth? Line 215, should rare be rarer? Lines 216 and 280 have comma splices. Line 236, awkward sentence. Line 243, varianthad should be corrected. Line 274 omit comma after diagnostic. Line 280, what is the meaning of the question?
Round 2
Reviewer 2 Report
The authors have improved their manuscript and presented their data in a more concise way. Still their work remains a case study and not a research work. Their work could have been amended with more data (as there are numerous studies and data freely available).
